# On the Effect of Bilateral Eye Movements on Memory Retrieval in Ageing and Dementia

**DOI:** 10.3390/brainsci12101299

**Published:** 2022-09-27

**Authors:** Megan Polden, Trevor J. Crawford

**Affiliations:** Psychology Department, Centre for Ageing Research, Lancaster University, Bailrigg, Lancaster LA1 4YF, UK

**Keywords:** bilateral eye movements, saccades, memory retrieval, word recognition, Alzheimer’s, mild cognitive impairment, Parkinson’s

## Abstract

It has been reported that performing bilateral eye movements for a short period can lead to an enhancement of memory retrieval and recall (termed the “saccade induced retrieval effect (SIRE)”). The source of this effect has been debated within the literature and the phenomenon has come under scrutiny as the robustness of the effect has recently been questioned. To date investigations of SIRE have largely been restricted to younger adult populations. Here, across two experiments, we assess the robustness and generalisability of the SIRE specifically in relation to disease and ageing. Experiment 1 employed a between subject’s design and presented younger and older participants with 36 words prior to completing one of three eye movement conditions (bilateral, antisaccade or a fixation eye movement). Participants then performed a word recognition task. Experiment 2 assessed the SIRE in individuals diagnosed with Alzheimer’s, Mild cognitive impairment and Parkinson’s by employing an online within subject’s design. Results showed no significant difference between groups in the number of words recognised based on eye movement condition. Neither experiment 1 or 2 replicated the SIRE effect therefore the findings from this study add to the growing number of studies that have failed to replicate the SIRE effect.

## 1. Introduction

Traditionally eye movements and memory have been studied in quite separate domains. There is now growing evidence of a close interaction between eye movements and working memory. For example, evidence derived from neuropsychological studies of people with dementia revealed that eye movements can be indicative of memory and cognitive impairment [1,2]. One line of research purports that eye movements interfere and disrupt working memory processes [3]. Pearson & Sahraie [4] across 5 experiments contrasting the effects of eye movements, limb movements and attention shifts on working memory, demonstrated a crucial role for oculomotor control processes during rehearsal of location representations in working memory. Later research assessing this claim more specifically revealed that it was eye movement attentional control processes (involved in retrieval, encoding or formation of images) and not the movement per se that produced disruptive effects and that these effects are limited to spatial working memory [5]. However, there is an emerging discord in understanding the relationship between eye movements and memory. The critical role of eye movement activity in spatial memory was highlighted using the abducted eye paradigm by Pearson et al. [6], although as the effects may include combined influences of both prospective planning and sensory representation of memory items [5]. Ryan and colleagues [7] stated that “eye movements may be functional for the formation, retrieval and reconstruction of memory” claiming a close interaction in which eye movements directly facilitate working memory [7]. It is suggested that gaze fixations during encoding processes is related to neural markers of memory formation and functional activity in the hippocampus, with the restriction of eye movements during encoding negatively impacting subsequent memory [8,9]. However, whereas voluntary eye movements were interruptive in the studies above [4,5], it has also been claimed that intrusive voluntary eye movements can actually lead to enhancements in memory processes [10]. This claim is critical as it suggests there is a potential for eye movements to facilitate working memory processes in both healthy and clinical populations.

Christman et al. [10] hypothesised that bilateral eye movements can lead to enhancements on subsequent memory and recall tasks due to increasing interhemispheric interaction which plays a role in episodic memory processes [11]. These bilateral eye movements incorporate the sequential gaze shift from left to right visual field in quick succession (Figure 1). They demonstrated that a sequence of bilateral eye movements for as little as 30 s can produce an enhancement effect on episodic memory specifically word recall. The so-called “saccade induced retrieval effect” (SIRE) has been replicated [12,13] and applied to various stimulus’s types such as autobiographic memory, spatial memory and episodic future thinking [14,15,16]. Although current literature has focused on assessing the SIRE effect across various stimuli, the effect has predominately been assessed in young adult populations and has not yet been applied to populations with cognitive deficits. Here, across two experiments we examined the SIRE effect in older adult populations and clinical populations with known cognitive and memory deficits (people with Alzheimer’s Disease, Mild cognitive impairment and Parkinson’s Disease) assessing the effect in relation to ageing and disease. 

Lyle and Martin [17] suggest that activation of the frontoparietal attention network could be the cause of the SIRE. If this is the case, attentional control tasks such as the antisaccade task, could elicit the SIRE effect. The antisaccade task presents participants with a single target and requires participants to shift their gaze and attentional focus to the opposite side. This task employs top-down control processes including inhibitory control, working memory and other executive operations. Regions of the frontoparietal network are activated during the task similar to episodic memory retrieval potentially making the task an effective priming method for a subsequent memory retrieval task. In the current study we assess the potential of antisaccade eye movements to enhance memory retrieval.

Experiment 1 examined antisaccadic (top-down control task) and bilateral eye movements and their ability to enhance word memory retrieval, utilising eye tracking to objectively monitor the eye movements. If the enhancement effect is specific to simple bilateral eye movements this will provide support for the interhemispheric interaction hypothesis [10] however if only the more complex anti saccadic eye movements are able to elicit the SIRE effect, then this would provide support for the attentional control hypothesis [17]. A more accurate understanding of the cause of the SIRE effect will help to produce a more reliable, robust, and replicable effect. As older adults are generally more susceptible to memory decline and reductions in cognitive processes [18,19], the effects of bilateral eye movements may increase with age. Younger adults may be more prone to ceiling effects on memory recall tasks and reduced ceiling effects as a result of memory decline in older adult populations may result in increased enhancements effects following bilateral eye movements. Due to this, the current study explored the SIRE in relation to ageing by including both healthy younger and older adults.

## 2. Materials and Methods

### 2.1. Participants

The study included 68 younger adults (mean age = 23.03, SD = 3.91, age range = 18–35 years) and 59 older adults (mean age = 63.55, SD = 6.71, age range = 55–90 years). The participants were white British or European fluent English speakers with a minimum of 11 years in formal education. The younger adults were recruited via the Lancaster University Research Participation System and the older adults were recruited from the local community. The younger and older adults were assigned to one of three experimental conditions: Bilateral prosaccadic eye movements, antisaccade eye movements and a fixation condition with no eye movements (Fixation condition). Participants were counterbalanced across the conditions. Only strongly right-handed individuals were included in the study due to research demonstrating inconsistent results of the SIRE effect in left-handed and ambidextrous individuals [17,20,21]. 

The following exclusion criteria was applied: previous head trauma, stroke, cardiovascular disease, physical or psychological conditions severe enough to affect their ability to participate, previous and current alcohol or substance misuse. Participants with focal cerebral lesions, history of neurological disorders (e.g., Parkinson’s disease, multiple sclerosis, epilepsy, amyotrophic lateral sclerosis, muscular dystrophy), neurodegenerative or cerebrovascular disease (including ischemic stroke, haemorrhagic stroke, atherosclerosis). 

The G*Power software version 3.1.9.7 was used to conduct a power analysis to determine the minimum sample size to ensure adequate power. The power level was set at 0.80 with an error of 0.05 for the analysis [22]. The effect size used (d = 0.495) was based on the Christman et al. [10] article for the comparison of horizontal eye movements to no eye movements. The analysis indicated a required sample size of 45 participants for a between subject’s design. A between subject’s design was used to replicate and maintain consistency with the Christman et al. [10] study. Therefore, we aimed to collect a minimum of 45 participants in experiment 1. All participants included in the study had normal or corrected to normal vision. Written informed consent was gained from all participants. Ethical approval was granted by Lancaster University Ethics committee in May 2018. 

### 2.2. Neuropsychological Assessments

The memory recognition task consisted of 72 words (see Appendix A) which were sourced from Friendly’s (1996) online word list generator consisting of 925 nouns collated by Paivio et al. [23] and scaled for print frequency, meaningfulness, imagery, concreteness. Two-word lists were created, a target and a foil word list both consisting of 36 words controlled for moderate meaningfulness, frequency, concreteness, and imagery scores. The words ranged from five to eight letters and included 2–4 syllables. The 36 target words were displayed via Microsoft PowerPoint, version 2013, singularly on the centre on the screen for 5 s per word automatically being replaced by the next word. Participants were informed that they would be asked to identify the words in a later task and asked to remember as many words as possible. Participants were later presented with a randomly mixed list of the total 72 words (36 target and 36 foil words) and were given 2 min to select the words they remembered. The memory recognition task produced two scores which are assessed here: correct words identified, and false (incorrect) words identified. An overall task score was calculated by subtracting the total number of false words identified from the total number of correct words identified for each participant. This measure allows for differences in task strategy to be controlled for. 

The digit and spatial span [24], forwards and reversed, were performed to assess working memory and acted as a distractor task to prevent rehearsal of the words. The Edinburgh Handedness Inventory [25] assessed handiness dominance. Participants scoring below >80 and not classed as strongly right-handed were excluded from the final analysis. 

### 2.3. Eye Movement Tasks

The study was a between factor design with participants randomly allocated to one of three eye movement conditions: bilateral prosaccade eye movement, antisaccade eye movement and a fixation condition. The Saccadometer Advanced software version A358 was used to record participants eye movements for the bilateral and antisaccade eye movement conditions to ensure compliance. Participants were seated 5 ft away from a plain white wall in which the lights from the Saccadometer were presented. A calibration trial was completed prior to the task. For the bilateral eye movement condition, participants were presented with a red target that moved from left to right repeatedly. The target moved every 500 ms resulting in two eye movements per second. Participants completed 70 trials lasting approximately 30–40 s.

For the antisaccade task, participants were presented with a central green fixation target presented for 100 ms followed by a single red target presented for 100 ms. There was a 100 ms gap between trials. Participants were instructed to avoid looking at the red target and instead look to the opposite side. Participants completed 40 antisaccade trials. The Saccadometer data was extracted and analysed using Latency Meter version 6.3. 

The fixation condition was presented using PowerPoint. The fixation condition consisted of a central red dot that flashed every 500 ms presented on a white background. The dot remained in the same central location throughout the 30 s eye movement display. The fixation eye movement was designed as a control condition and provided the flashing stimulation without the bilateral movement. Participants were instructed to maintain their gaze on the presented target as it flashed at the centre of the screen for 30 s. Compliance was monitored visually by the experimenter due to the absence of an eye movement for this condition.

## 3. Results

A multivariate ANOVA was conducted investigating the effect of eye movement condition and ageing on correct and false words identified. A total of 11 people from the younger group and 1 person from the older group were excluded from the analysis due to scoring below 80 on the Edinburgh Handedness Inventory [25]. No participants were removed from the analysis due to poor compliance with the eye movement task. 

### 3.1. Memory Assessments

An ageing effect was found on the spatial span task (total score), F(1,113) = 5.097, *p* = 0.026, with younger participants recalling longer spatial patterns (Table 1). There were no significant differences in digit span task score (total score), F(1,113) = 0.011, *p* = 0.916. 

### 3.2. Correct Words Identified

Results showed that there was no significant effect of eye movement condition on the number of correct words identified on the recognition task, for the younger (F(2,54) = 1.66, *p* = 0.20, partial η^2^ = 0.58) or older adult group (F(2,55) = 0.099, *p* = 0.91, partial η^2^ = 0.004) (Table 2). Results showed no significant difference in the number of correct words identified based on age group, F(1,113) = 1.25, *p* = 0.27, partial η^2^ = 0.011. 

### 3.3. False Words Identified 

Results showed no significant effect of eye movement condition on false word recognition for the younger adult group, F(2,54) = 0.26, *p* = 0.77, partial η^2^ = 0.09 or the older adults group, F(2,55) = 0.023, *p* = 0.98, partial η^2^ = 0.001 (Table 3). There was a significant ageing effect on the number of false words identified, (F(1,113) = 5.66, *p* = 0.019, partial η^2^ = 0.048) with older participants (M = 5.55, SD = 4.14) identifying significantly more false words than the younger participant group (M = 3.88, SD = 3.37). This indicates a performance ageing effect or a difference in task strategy across the groups. 

### 3.4. Task Score Words

Participant task score was calculated by deducting the number of false words from the number of correct words identified for each participant. This was to control for strategy differences. There was no significant effect on task score due to the eye movement condition for the younger (F(2,54) = 1.47, *p* = 0.24, partial η^2^ = 0.052) or older adults (F(2,55) = 0.057, *p* = 0.95, partial η^2^ = 0.002) (Table 4). 

Results showed that there was a significant effect of age on memory task scores, F(1,113) = 5.24, *p* = 0.024, partial η^2^ = 0.044. Older participants (M = 18.66, SD = 7.09) scored significantly lower than younger adult participants (M = 21.51, SD = 6.24). This indicates a potential strategy difference between older and younger participants. Older participants select a higher number of false words impacting on the overall task score. Younger participants may be more reserved in their selections and therefore select fewer false words. 

## 4. Discussion

Bilateral eye movements have previously been shown to elicit memory retrieval enhancement effects in younger adult populations. The cause of the SIRE effect has been debated in the current literature with conflicting accounts. The current study assessed the effect of bilateral and antisaccade eye movements on the number of words younger and older participants were able to recognise. Results from the current study did not replicate the enhancement effect of bilateral eye movements on word memory recognition and recall [10,17]. The effect was not found in younger or older adults indicating that the effect may not be as robust as previously evidenced in the literature. Results showed that neither bilateral eye movements nor antisaccadic eye movements produced a performance enhancement on the word recognition task. There was no significant difference between the groups in the number for correct words identified or false words identified. This indicates that bilateral stimulation or performing a top-down control task (antisaccade task) may be ineffective at producing an enhancement effect on a word recognition task in healthy younger and older adults. 

An ageing effect was found when assessing word recognition task score, with younger adults producing higher task scores and identifying less false words than older adults. This indicates an age-related deterioration in memory recognition and recall capabilities. It is also possible that this result is due to a strategy variation between the age groups. Younger adults may be more conservative when selecting words and therefore may only select words they are more certain are correct resulting in the selection of less false words. However, if older adults are less reserved in their judgements, they may select a greater number of words or be more inclined to guess on the task. 

In recent years, there has been a growing literature that has failed to replicate the SIRE effect, bringing into question the robustness and replicability of the effect. Matzke et al. [26] conducted a preregistered study with the aim to replicate the enhancement effect originally found in the Christman et al. [10] study. After reviewing the literature, they attempted to design an optimum research design to investigate the robustness of the effect and used Bayesian statistics to analyse the results. Results revealed no significant variations in the number of words recognised depending on the eye movements condition leading to questions surrounding the robustness of the effect and the conditions in which the effect is optimised. The researchers suggested that previous significant results could be due to the use of *p* values rather than Bayesian statistics which are arguably a more stringent technique [27]. However, many studies displaying the effect have yielded robust *p* values [10,12,17]. Roberts et al. [28] conducted two experiments aiming to replicate the SIRE effect. In their first experiment they successfully replicated the SIRE effect found in the Christman et al. [10] study, however results showed weak Bayesian evidence indicating weak support for the experimental effect found. However, their second experiment that expanded the sample size and assessed vertical and horizontal saccades separately failed to replicate the effect. It was concluded that the SIRE effect is prone to inconsistencies and is very sensitive to experimental design. This study further highlights the inconsistencies and apparent lack of robustness of the SIRE effect. Nevertheless, the failure to replicate should not diminish the extensive literature that has supported this effect and further research is needed to examine the effects systematically. 

A recent systematic review [29] assessing the SIRE effect in relation to horizontal and vertical saccades reported that across 22 studies there was a significant facilitation of horizontal on memory performance providing strong evidence for the SIRE effect. There was no significant effect of vertical saccades on memory retrieval performance and found that handedness influenced the effect with strongly right-handed individuals benefiting more from horizontal saccades than inconsistent handers. These results provide support for the interhemispheric interaction hypothesis and demonstrates the potential for horizontal saccades to enhance memory performance. This systematic review provides strong evidence for the SIRE effect across multiple studies and indicates the effects validity. However, it should be noted that systematic reviews are highly susceptible to publication bias that may have influenced this result. Future research and replications are required to establish the validity and replicability of the SIRE effect. Research should also be conducted with wider populations and participant samples of various ages, ethnicities and clinical groups to investigate the generalisability of the effect.

## 5. Experiment 2

Experiment 2 expanded on experiment 1 by investigating the potential of bilateral eye movements in people with memory impairments and neurodegenerative disease. Bilateral eye movements have previously demonstrated an enhancement effect in neurotypical adults however current literature has not investigated the effect in people with mild cognitive impairment (MCI) or dementia due to Alzheimer’s. Disease (AD). AD is one of the most common causes of Dementia and is a prominent neurodegenerative disease [30]. People with AD often show reduced episodic memory, attentional control and executive functioning [31]. Due to the reduced episodic memory, AD participants may show greater benefit from the SIRE effect and enhanced susceptibility. Further, research has shown that people with AD often display abnormal eye movements on tasks such as pro and antisaccade tasks [1,32]. Cognitive impairment is also well-recognised in Parkinson’s (PD) and similar to AD populations display abnormal eye movements on pro and antisaccade tasks [33,34,35]. Due to eye movement variations on well-established paradigms, it cannot be assumed that the previously found SIRE effect will generalise. Therefore, assessing the SIRE effect in people with AD and MCI is important to establish the robustness of the effect in populations with reduced memory capabilities and the potential benefits of the SIRE effect for these populations. 

Although experiment 1 did not yield the enhancement effect, this may have been due to the highly educated sample used leading to a ceiling effects on memory performance. The potential therapeutic benefits of the SIRE may not be fully known due to the lack of research in populations with reduced memory capabilities. The reduced memory recognition and recall capabilities in people with Alzheimer’s and mild cognitive impairment may facilitate the enhancement effect due to lower baseline memory capabilities. People with Alzheimer’s disease experience memory and executive functioning deficits and techniques to aid and enhance memory capabilities which even temporarily could have a great impact on everyday life and activities. 

Therefore, the current experiment compared bilateral eye movements against an eye fixation movement condition in both older healthy adults and people with Alzheimer’s disease and mild cognitive impairment and in people with Parkinson’s disease, a population who often experience motor deficits alongside cognitive deficits. Assessing the SIRE in clinical populations will allow investigation into potential therapeutic benefits from bilateral eye movements.

Due to the COVID-19 pandemic restricting face to face testing, experiment 2 was converted to an online study using the online testing tool Gorilla. Given the high level of compliance with the eye movement tasks in experiment 1, eye tracking was not implemented in experiment 2.

## 6. Materials and Methods

### 6.1. Participants

Experiment 2 included 27 Healthy older adults (Mean age = 69.74 years, SD = 7.57 years), 10 participants with Dementia due to Alzheimer’s Disease or mild cognitive impairment (Mean age = 75.6 years, SD = 5.10 years), and 31 participants with Parkinson’s Disease (Mean age = 64.35 years, SD = 7.95 years). The inclusion and exclusion criteria and recruitment strategy were kept consistent with experiment 1 for healthy older controls.

The AD and MCI participants were recruited via various National Health Trusts and memory clinics in the UK who distributed the online task. Participants had previously received a clinical diagnosis following a full neurocognitive assessment with a dementia specialist. The AD participants met the requirements for the American Psychiatric Association’s Diagnostic and Statistical Manual of Mental Disorders (DSM IV) and the National Institute of Neurological and Communicative Disorders and Stroke (NINCDS) for AD. 

The MCI participants had received a diagnosis of dementia due to mild cognitive impairment and met the following criteria [36]: (1) subjective reports of memory decline (reported by individual or caregiver/informant); (2) memory and/or cognitive impairment (scores on standard cognitive tests were >1.5 SDs below age norms); (3) Activities of daily living were moderately preserved. The inclusion and exclusion criteria and recruitment methods for the older adults was consistent with experiment 1. Ethical approval was granted by Lancaster University Ethics committee and by the NHS Health Research Authority, Greater Manchester West Research Ethics Committee. 

Participants with Parkinson’s Disease had received a formal diagnosis of Parkinson’s Disease and were recruited through the local community and Parkinson’s UK database. All PD participants were receiving parkinsonian medication and completed the study while under their usual medication regime. The Movement Disorder Society–Unified Parkinson’s Disease Rating Scale (MDS-UPDRS) [37] was used to assess Parkinsonian symptoms. Of the 31 PD participants the average length of time since diagnosis was 5 years and 4 months. All 31 participants with Parkinson’s were receiving Parkinsonian medication and were tested under their normal medication regime. Fifteen participants were taking a dopamine agonist (e.g., ropinirole), 11 participants were taking combination (containing levodopa and a peripheral dopadecarboxylase inhibitor, e.g., Madopar), 11 were taking a monoamine oxidase inhibitor (e.g., rasagiline) and 2 patients were taking a catechol-O-methyl transferase inhibitor (e.g., entacapone).

Consistent with experiment 1, G*Power software version 3.1.9.7 was used to conduct a power analysis with the power level set at 0.80 and an error of 0.05. The effect size used was d = 0.495 based on the Christman et al. [10] article. The analysis indicated a required sample size of 33 participants for a within-subjects design. 

### 6.2. Memory Assessments

For experiment 2, a within subject design was employed to control for the variability of memory abilities in AD and MCI participants. A within study design was employed due to the increase variability in patient groups. The study was created and controlled via the online testing tool Gorilla Experiment Builder (www.gorilla.sc, accessed on 21 September 2022). Due to the within study design a further word list was created. The criteria and procedure for creating the second word list was consistent with experiment 1. The presentation of the eye movement conditions, and the word lists were counterbalanced across the groups. Participants were randomly assigned to one of four sequences for completing the study. Participants were provided with two links, to access the online experiment and a Zoom call with the researcher. The researcher remained present on the call while participants completed the study to ensure understanding and compliance with the study and eye movement tasks. The Montreal Cognitive Assessment (MOCA) [38] was completed to indicate probable dementia. Scores below 26 are indicative of MCI and scores below 21, indicative of AD. The MoCA was completed verbally with the experiment. 

### 6.3. Online Tasks

Participants accessed the experiment task via a URL link sent to the participant. The Edinburgh Handedness Inventory [25] was completed with the procedure consistent with experiment 1. The subjective memory complaints questionnaire (SMCQ) was performed to assess the participants perception of their memory impairment [39]. The SMCQ consisted of 14 questions in which the participant responded either yes or no, for example “Do you have difficulty in remembering a recent event?”. 

The word memory task was converted to an online version with the procedure kept consistent with experiment 1. Participants were presented 36 words that they were instructed to remember and recognise in a later task. Each word was shown on the screen individually for 5 s and automatically changed to the next word consistent with experiment 1. The digit span task [24] was converted to an online version in which each number in the sequence would be individually displayed on the screen for 3 s before changing to the next number in the sequence. Participants were then presented with an entry box where they were instructed to type the number sequence, they had been shown using their keyboard. This procedure was used for both the forwards and backwards version of the task. The digit span task acted as a baseline memory assessment but also to prevent rehearsal of the words prior to the recognition tasks. Following the completion of the assigned eye movement, participants completed a word recognition task consistent with experiment 1. Participants were presented with 72 words (36 target words and 36 false words) and asked to select the words they could recognise from the previous presentation. The same procedure was then repeated for the second word list and eye movement (Figure 2). There was a 5 min delay period between conditions to avoid carry-over effects [40]. 

### 6.4. Eye Movements 

The eye movement task included two conditions: Eye fixation and bilateral eye movement. Participants were instructed to sit 55 cm away from the computer screen to maintain a consistent visual angle.

For the bilateral condition participants were presented with a central cross fixation for 250 ms to centre their eye prior the start of the task. A red dot then flashed from the left side of the screen to the right side every 500 ms creating two eye movements per second for 60 trials. The distance of the red target from the central fixation point was 5 cm. The red dot measured 15 mm in diameter (visual angle, 1.56). Participants were instructed to follow the red dot with their eyes as accurately as possible. For the fixation eye movement condition, the fixation was displayed at the centre of the screen for 250 ms. This was replaced by a red dot that flashed at the centre of the screen repeatedly, once every 500 ms. Participants were instructed to maintain their gaze on the target as it flashed in the centre of the screen for 30 s. Participants completed 60 trials lasting approximately 30 s. To monitor compliance with the eye movements, the researchers visually observed the participant completing the eye movements via their webcam.

## 7. Results

The data was extracted from the online testing tool Gorilla and analysed using SPSS version 27. A multivariate ANOVA was conducted investigating performance variation and potential bilateral enhancement effects on the groups. Group variations between the number of correct and number of false words identified, and overall task score was compared using an ANOVA analysis. For the analysis, participants with AD and MCI were combined into a cognitively impaired group (CI) due to the small sample sizes of these groups individually. Three participants from the older control group and three participants from the Parkinson’s group were excluded from the analysis due to scoring <80 on the Edinburgh Handedness Inventory [25] indicating inconsistent handedness. No participants were removed from the analysis due to poor compliance with the eye movement tasks. 

### 7.1. Memory Assessments

Results revealed a significant group effect on MOCA task score, (F(2,65) = 17.82, *p* < 0.0001) with the CI group producing significantly lower task scores compared to the PD and OC group (Table 5). The PD group produced significantly lower task scores compared to the OC group on the MOCA. For the digit span task, significant groups effects were found on both the forwards F(2,65) = 21.86, *p* = 0.023) and backwards F(2,65) = 37.87, *p* = 0.001) versions of the task. For the forwards digit span, the PD group scored significantly lower on the task compared to the OC group. There were no significant differences between the other participant groups. For the backwards version of the task, results revealed that the CI group and the PD group scored significantly lower than the control group. There was a significant group effect on the SMCQ reporting (F(2,65) = 79.34, *p* <.0001) with expectedly the CI group reporting more subjective memory impairments compared to the PD and the OC group. 

### 7.2. Correct Word Identified 

When assessing the effect of eye movement condition across the groups on the number of correct words identified, results revealed no significant main effects for participant group (F(2,65) = 2.11, *p* = 0.130, partial η^2^ = 0.061) or eye movement condition (F(1,65) = 1.08, *p* = 0.303, partial η^2^ = 0.016). There were no significant interaction effects between participant group and eye movement condition (F(2,65) = 0.772, *p* = 0.466, partial η^2^ = 0.023) (Table 6). 

#### 7.2.1. Participants with Cognitive Impairment 

The effects of the eye movement condition (bilateral vs. Fixation) may affect the individual groups differently or to varying extents. Due to this we assessed the effect of the eye movement condition separately for each participant group. Although the AD and MCI participants recognised a larger number of words during the bilateral condition compared to the fixation condition the change was not significant, F(1,9) = 1.549, *p* = 0.245, partial η^2^ = 0.147. This non-significant result may be due to a lack of power due to the small sample size.

#### 7.2.2. Participants with Parkinson’s Disease

Results revealed no significant effect of eye movement condition on the number of correct words recognised for the Parkinson’s group, F(1,30) = 0.109, *p* = 0.743, partial η^2^ = 0.004.

#### 7.2.3. Older Control Participants

There was no significant difference in the number of correct words recognised as a result of eye movement condition for the control group, F(1,26) = 0.030, *p* = 0.864, partial η^2^ = 0.001. 

### 7.3. False Word Identified

When assessing the effects of eye movement condition and participant group on the number of false words participants recognised, results revealed no significant effect of eye movement condition (F(1,65) = 1.89, *p* = 0.174, partial η^2^ = 0.028) or participant group (F(2,65) = 0.941, *p* = 0.395, partial η^2^ = 0.028). No significant interactions were found, F(2,65) = 0.924, *p* = 0.402, partial η^2^ = 0.028 (Table 7). 

#### 7.3.1. Participants with Cognitive Impairment

AD and MCI participants recognised a higher number of false words during the bilateral condition compared to the fixation eye movement condition, however this difference was not significant, F(1,9) = 1.340, *p* = 0.277, partial η^2^ = 0.130. 

#### 7.3.2. Participants with Parkinson’s Disease

There was no significant effect of eye movement condition on the number of false words recognised for the PD group, F(1,30) = 0.084, *p* = 0.773, partial η^2^ = 0.003.

#### 7.3.3. Older Control Participants

Similar to the AD, MCI and Parkinson’s participants, the older controls showed no significant effect of eye movement condition on the number of false words recognised, F(1,26) = 1.013, *p* = 0.323, partial η^2^ = 0.038. 

### 7.4. Task Score 

Task score was calculated by subtracting the number of false words identified from the number of correct words recognised. This was to adjust for variations in strategy across participants. Consistent with the results for the correct and false words identified, there was no significant effect of eye movement condition (F(1,65) = 0.133, *p* = 0.717, partial η^2^ = 0.002) or participant group (F(2,65) = 2.45, *p* = 0.094, partial η^2^ = 0.071) on task score. There were no significant intervention effects, F(2,65) = 0.572, *p* = 0.567, partial η^2^ = 0.018 (Table 8). 

#### 7.4.1. Participants with Cognitive Impairment 

Results showed no significant difference between task score between the bilateral and fixation eye movement conditions, F(1,9) = 0.226, *p* = 0.646, partial η^2^ = 0.025. 

#### 7.4.2. Participants with Parkinson’s Disease

There was no significant effect of eye movement condition on task score for the PD group, F(1,30) = 0.826, *p* = 0.371, partial η^2^ = 0.028. 

#### 7.4.3. Older Control Participants

Results showed that there was no significant difference in task score based on eye movement condition for the older control participants, F(1,26) = 0.298, *p* = 0.590, partial η^2^ = 0.011.

## 8. Discussion

Experiment 2 investigated the SIRE effect in populations with memory impairments and in healthy older adult populations. Previous research has only investigated the SIRE effect in healthy older and younger adult populations [10,17] and has failed to assess the potential of the SIRE effect in populations with cognitive impairments. The benefits and effects of bilateral eye movements could have even greater benefit to disease populations and elicit stronger retrieval enhancements. Results from experiment 2 which assessed the SIRE effect in people with AD, MCI, PD and healthy older adults failed to replicate the SIRE effect. There was no significant difference in the number of correct or false words recognised across the participants groups based on the eye movement condition. Conducting bilateral eye movements failed to induce an enhancement in memory retrieval in AD, MCI, PD and older adult populations. This result was consistent with experiment 1 that also failed to replicate the SIRE effect in younger and older populations. These results add to the growing amount of literature that has failed to replicate the SIRE effect [26,28] and provides new knowledge that the SIRE effect may not be beneficial in populations with memory impairments and may not generalise to clinical populations. The inconsistent findings and the growing number of failures to replicate undermines the robustness of the effect and its potential as a clinical tool. 

Research involving the SIRE effect have used a variety of methodologies with one of the main variations being between vs. within study designs. Here, experiment 1, employed a between study design and experiment 2 employed a within study design due to the increased variability in cognitive abilities in AD, MCI and PD populations. The majority of SIRE literature has used a between-subjects design although studies have successfully demonstrated the SIRE effect in within-subjects designs [40]. Brunye et al. [40] employed a 10 min delay between conditions and did not report finding carry-over effects. Additionally, Roberts et al. [28] used a five-minute delay between conditions and although did not find the SIRE effect, they did not find any reliable order effects indicating that carry-over effects were unlikely. Based on previous literature using a within-subjects design as opposed to a between-subjects design, should not significantly influence or diminish the effect. Further due to a reduced overall variance, particularly high in clinical populations, within-subjects designs could even increase the likelihood of finding an effect providing carry-over and order effects are controlled for. Here, the SIRE effect was unable to be replicated with either a between subject or a within-subjects design indicating that the lack of an enhancement effect found in this study was unlikely a result of subject design variations. 

It should be noted that although experiment 1 and 2 closely mimicked previous experimental designs [10], they were not direct replications due to the attempt to expand on existing literature. Methodological variations employed in this study may have weakened or diminished the effect for example the addition of the antisaccade condition and eye tracking in experiment 1 and the shift to an online study with a within study design in experiment 2. Recent evidence [28] suggests that the SIRE effect is highly sensitive to experimental methodology and only appears to be present in specific conditions. The effect appears to only be present in people with certain handedness conditions (strong right-handers), it may be more effective when between methodologies are employed and may only be present in certain lab conditions due to slight variations in methodologies. With so many factors that weaken or diminish the effect, it is clear that the effect lacks reliability and stability. Experiment 2 was conducted online and although there are benefits to online studies, certain methodological parameters can be more difficult to control, such as computer screen size and distance seated from the screen. Research suggests that the SIRE effect may lack robustness and stability and may not lend itself well to being tested in an online setting due to greater potential variations in experiment equipment and set up and less stringent experimental methods. A lab setting can employ a higher level of control and consistency and therefore may increase the likelihood of replicating the SIRE effect. However, here we accessed the SIRE effect in both a lab and online setting and failed to replicate the effect. Repeated failures to replicate can often indicate a nuance in participant demographics or methodology that has not been identified or considered. These nuances could be integral to the effect and until specific boundary conditions are clearly specified, the effect may continue to lack robustness. Direct replications should be conducted to assess the robustness of the effect and to document specific boundary conditions of the effect. 

It should be considered that certain individuals may be more susceptible to enhancement effects than others and methodologies that look at enhancement effects on an individual level may show more promise [41]. It is clear from the literature that the effect is more evident in strongly right-handed individuals [10] and it is possible that other unknown characteristics may mediate the effect. 

To date, research has not examined the potential enhancement effect on memory retrial when employing multimodal stimuli rather not solely oculomotor involvement. Marandi et al. [41] demonstrated that auditory input potentially improves the visual recall process. Combining enhancement effects (e.g., oculomotor and auditory techniques) in a multimodal stimuli approach may result in greater enhancement effects on memory retrieval creating a more robust and replicable effect. Future research should examine enhancement effects on memory recall and retrieval by employing a multimodal stimuli approach and examining the effect on an individual level rather than population level. 

Across the literature, multiple tasks and stimulus types have been employed when assessing the enhancement effects of eye movements on cognition with mixed results [10,13,42,43]. Future research should examine enhancement effects on memory recall and retrieval by employing a multi-modal stimuli approach and examining the effect of stimulus variables, such as emotional valence. Additional research is also required to examine the SIRE effect on multiple task types and stimuli due to the vast about of literature employing different methodologies.

It is well known that clinical populations are often more variable than the general population. Effects that lack robustness can be even more unstable in clinical populations. Due to this, the clinical applications of the SIRE effect should be questioned. The current study failed to demonstrate the SIRE effect in populations with memory and cognitive impairments and fails to provide evidence for bilateral eye movements to have therapeutic benefits for AD, MCI or PD populations. However, due to the small sample sizes in experiment 2, particularly for the AD and MCI group, clinical application should not be ruled out. The study may have lacked sufficient power, and this could likely have caused the null result observed in experiment 2. The cognitive impairment group included only 10 participants and power analysis indicating a sample size of 33 (approximately 11 per group) to produce a significant result at the *p* < 0.05 level. The AD and MCI group recognised more correct words during the bilateral conditions compared with the no eye movement condition although not significant. This null result could be due to a lack of power from low participant samples and therefore results from experiment 2 should not rule out the potential benefits of the SIRE effect in clinical populations. Future research should continue to assess the SIRE effect in people with AD, MCI and PD with adequate sample size to achieve sufficient power. Although, prior to clinical applications, consistent replications are required to demonstrate a stable and robust effect with clear boundary conditions. 

## 9. Conclusions

Future research should focus on establishing the robustness of the SIRE effect by performing direct pre-registered replications to validate the effect. Research should aim to establish clear and precise boundary conditions in which the effect is present, robust, and replicable. Such replications could provide a deeper understanding of the literature and findings could help re-examine and enhance existing theories.

## Figures and Tables

**Figure 1 brainsci-12-01299-f001:**
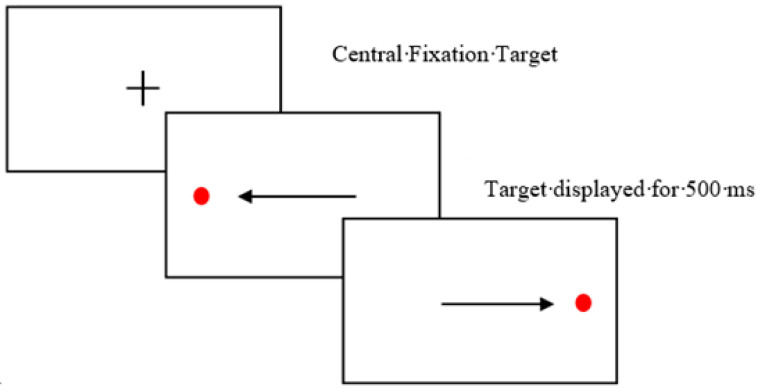
Bilateral eye movement task. Red target flashes from the left side of the screen to the right repeatedly for 30 s. The target moves every 500 ms resulting in 2 horizontal saccades per second. Arrows indicate the direction of the horizontal saccade.

**Figure 2 brainsci-12-01299-f002:**
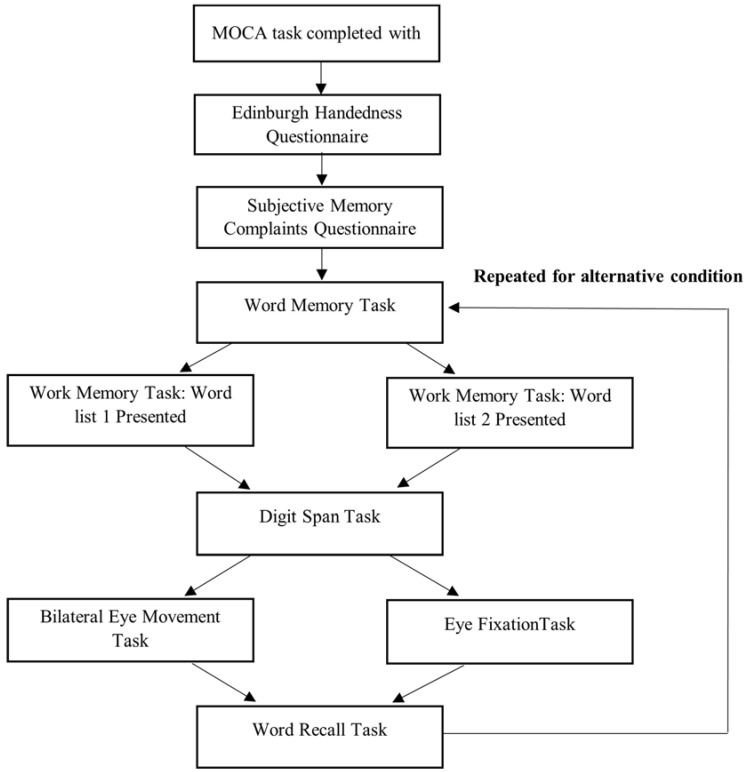
Study Flow Diagram for Experiment 2.

**Table 1 brainsci-12-01299-t001:** Means and standard deviations for the neurological assessments.

	Digit Span Forward		Digit Span Backwards		Digit Span Total		Spatial Span Forwards		Spatial Span Backwards		Spatial Span Total	
	M	SD	M	SD	M	SD	M	SD	M	SD	M	SD
Young Adults (n = 57)	12.19	2.29	8.14	2.66	20.33	4.12	8.17	2.70	7.35	1.49	14.88	3.38
Older Adults (n = 58)	12.48	2.13	8.10	2.75	20.41	4.05	7.12	1.69	6.47	1.35	13.59	2.72

Note. Dependent variable: Task score.

**Table 2 brainsci-12-01299-t002:** Mean and standard deviations for the correct words identified following the eye movement conditions.

	Bilateral		Antisaccade		Fixation	
	M	SD	M	SD	M	SD
Young Adult group (n = 57)	26.70	5.66	25.79	5.44	23.50	5.49
Older Adult group (n = 58)	23.90	5.47	24.05	6.88	24.21	5.70

Note. Dependent variable: Number of words recognised.

**Table 3 brainsci-12-01299-t003:** Mean and standard deviations for the false words identified following the eye movement conditions.

	Bilateral		Antisaccade		Fixation	
	M	SD	M	SD	M	SD
Young Adult group (n = 57)	4.15	3.72	3.42	3.06	4.06	3.40
Older Adult group (n = 58)	5.40	5.18	5.68	3.35	5.58	4.14

Note. Dependent variable: Number of false words recognised.

**Table 4 brainsci-12-01299-t004:** Mean and standard deviations for task score following the eye movement conditions.

	Bilateral		Antisaccade		Fixation	
	M	SD	M	SD	M	SD
Young Adult group (n = 57)	22.55	6.71	22.37	6.19	19.44	5.55
Older Adult group (n = 58)	18.50	7.80	18.37	6.95	19.10	6.80

Note. Dependent variable: Task score.

**Table 5 brainsci-12-01299-t005:** Mean and standard deviations and group post hoc comparisons for the cognitive assessments.

	Participants with Cognitive Impairments (n = 10)		Parkinson’s Group (n = 31)		Older Control Group (n = 27)		Post Hoc Comparisons		
							CI vs. PD	CI vs. OC	PD vs. OC
	M	SD	M	SD	M	SD			
MOCA Task Score	22.40	4.95	25.26	2.19	28.04	2.03	0.013 *	<0.001 *	0.001 *
Digit Span Forward	9.50	2.22	9.61	2.09	11.22	2.62	0.990	0.122	0.029 *
Digit Span Backwards	6.50	1.72	7.22	1.94	9.15	2.44	619	0.004 *	0.003 *
SMCQ score	7.40	3.50	3.87	3.10	2.74	2.31	0.004 *	<0.001 *	0.301

Note. Dependent variable: Task score. CI—Participants with cognitive impairment; OC—older control group; PD—Parkinson’s group. * Significant at *p* < 0.05 level.

**Table 6 brainsci-12-01299-t006:** Mean and standard deviations and group post hoc comparisons for the correct words identified following the eye movement conditions.

	Participants with Cognitive Impairments (n = 10)		Parkinson’s Group (n = 31)		Older Control Group (n = 27)		Post Hoc Comparisons		
							CI vs. PD	CI vs. OC	PD vs. OC
	M	SD	M	SD	M	SD			
Bilateral Eye Movement	20.10	9.15	24.03	6.79	23.26	7.97	0.339	0.507	0.922
Fixation Eye Movement	17.30	11.80	23.71	7.21	23.52	7.35	0.081	0.101	0.996

Note. Dependent variable: Correct words identified. CI—Participants with cognitive impairment; OC-older control group; PD—Parkinson’s group.

**Table 7 brainsci-12-01299-t007:** Mean and standard deviations and group post hoc comparisons for the false words identified following the eye movement conditions.

	Participants with Cognitive Impairment (n= 10)		Parkinson’s Group (n = 31)		Older Control Group (n = 27)		Post Hoc Comparisons		
							CI vs. PD	CI vs. OC	PD vs. OC
	M	SD	M	SD	M	SD			
Bilateral Eye Movement	7.30	4.47	5.03	5.16	4.74	5.12	0.438	0.364	0.974
Fixation Eye Movement	5.60	4.35	5.23	5.28	3.81	3.87	0.973	0.555	0.484

Note. Dependent variable: False words identified. CI–Participants with cognitive impairment; OC—older control group; PD—Parkinson’s group.

**Table 8 brainsci-12-01299-t008:** Mean and standard deviations and group post hoc comparisons for task score following the eye movement conditions.

	Participants with Cognitive Impairments (n= 10)		Parkinson’s Group (n = 31)		Older Control Group (n = 27)		Post Hoc Comparisons		
							CI vs. PD	CI vs. OC	PD vs. OC
	M	SD	M	SD	M	SD			
Bilateral Eye Movement	12.80	5.79	18.22	10.61	18.51	10.50	0.304	0.280	0.993
Fixation Eye Movement	11.70	8.82	17.23	11.1	19.70	8.73	0.283	0.081	0.616

Note. Dependent variable: Task score. CI—Participants with cognitive impairment; OC—older control group; PD—Parkinson’s group.

## Data Availability

Data is available on open science framework: https://osf.io/8rpwg/ (accessed on 28 August 2022).

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
