# Peer review of "On the Effect of Bilateral Eye Movements on Memory Retrieval in Ageing and Dementia"

_brainsci, 2022, doi:10.3390/brainsci12101299_

Round 1
Reviewer 1 Report
This is a very interesting study examining the effect of bilateral eye movements of memory retrieval in both ageing and patient populations such as AD and PD.
It is a well written paper which addresses an important area, that the authors rightly say has been debated within the literature. This work therefore constitutes further evidence which will be of great interest to the scientific community.
I have a few minor comments for the authors:
(1) Were all the patients medicated? For instance, were there any unmedicated Parkinsonian patients?
(2) The authors might want to include the following papers as prior literature on antisaccades and a Parkinson's cohort.
Antisaccades and executive dysfunction in early drug-naive Parkinson's disease: The discovery study.Antoniades CA, Demeyere N, Kennard C, Humphreys GW, Hu MT. Mov Disord. 2015 May;30(6):843-7. doi: 10.1002/mds.26134. Epub 2015 Jan 20.PMID: 25600361
Oculomotor deficits in Parkinson's disease: Increasing sensitivity using multivariate approaches.Bredemeyer O, Patel S, FitzGerald JJ, Antoniades CA.
Front Digit Health. 2022 Aug 4;4:939677. doi: 10.3389/fdgth.2022.939677. eCollection 2022.PMID: 35990016
The effect of levodopa on saccades - Oxford Quantification in Parkinsonism study.
Lu Z, Buchanan T, Kennard C, FitzGerald JJ, Antoniades CA. Parkinsonism Relat Disord. 2019 Nov;68:49-56. doi: 10.1016/j.parkreldis.2019.09.029. Epub 2019 Sep 27.
PMID: 31621619
Author Response
Dear Reviewer,
Thank you for your helpful comments that have helped to improve the manuscript. Below are our responses to your comments. For your convenience, we have copied the changed sections here, and also in the manuscript where the revised sections are highlighted in yellow.
Reviewer 1
Reviewer Comment 1:
- Were all the patients medicated? For instance, were there any unmedicated Parkinsonian patients?
Author Response:
Thank you for this comment. All 31 participants with Parkinson’s were receiving parkinsonian medication and were tested under their normal medication regime. We have added further details to the manuscript regarding the medication regime of the Parkinsonian participants. The following paragraph has been added to the manuscript on page 8.
“All 31 participants with Parkinson’s were receiving Parkinsonian medication and were tested under their normal medication regime. Fifteen participants were taking a dopamine agonist (e.g., ropinirole), 11 participants were taking combination drugs (containing levodopa and a peripheral dopadecarboxylase inhibitor, e.g., Madopar), 11 were taking a monoamine oxidase inhibitor (e.g., rasagiline) and 2 patients were taking a catechol-O-methyl transferase inhibitor (e.g., entacapone).”
Reviewer Comment 2:
- The authors might want to include the following papers as prior literature on antisaccades and a Parkinson's cohort.
Antisaccades and executive dysfunction in early drug-naive Parkinson's disease: The discovery study.Antoniades CA, Demeyere N, Kennard C, Humphreys GW, Hu MT. Mov Disord. 2015 May;30(6):843-7. doi: 10.1002/mds.26134. Epub 2015 Jan 20.PMID: 25600361
Oculomotor deficits in Parkinson's disease: Increasing sensitivity using multivariate approaches.Bredemeyer O, Patel S, FitzGerald JJ, Antoniades CA.
Front Digit Health. 2022 Aug 4;4:939677. doi: 10.3389/fdgth.2022.939677. eCollection 2022.PMID: 35990016
The effect of levodopa on saccades - Oxford Quantification in Parkinsonism study.
Lu Z, Buchanan T, Kennard C, FitzGerald JJ, Antoniades CA. Parkinsonism Relat Disord. 2019 Nov;68:49-56. doi: 10.1016/j.parkreldis.2019.09.029. Epub 2019 Sep 27.
PMID: 31621619
Author Response:
Thank you for bringing our attention to this relevant literature. We have added these papers to our experiment 2 introduction section on page 7 with the following statement. “Cognitive impairment is also well-recognised in Parkinson’s (PD) and similar to AD populations display abnormal eye movements on pro and antisaccade task [33-35].”
Thank you again for your helpful comments aiding us to improve the manuscript.
Yours Sincerely,
Megan Polden and Professor Trevor J. Crawford

Reviewer 2 Report
The study provides evidence on robustness of bilateral eye movements for memory retrieval. There should be more discussion on why this method may or may not work on different people. For example, authors could refer to the following studies explaining their insights on how future work could enhance the memory retrieval process by adopting multimodal stimuli rather than only oculomotor involvement:
Marandi, R.Z. and Sabzpoushan, S.H., 2014. Using eye movement analysis to study auditory effects on visual memory recall. Basic and clinical neuroscience, 5(1), p.55.
or to what extent the lack of evidence could be related to the type of task for example
Leer, A., Engelhard, I.M., Lenaert, B., Struyf, D., Vervliet, B. and Hermans, D., 2017. Eye movement during recall reduces objective memory performance: An extended replication. Behaviour Research and Therapy, 92, pp.94-105.
Marandi, R.Z. and Gazerani, P., 2019. Aging and eye tracking: in the quest for objective biomarkers. Future Neurology, 14(4), p.FNL33.
Wynn, J.S., Shen, K. and Ryan, J.D., 2019. Eye movements actively reinstate spatiotemporal mnemonic content. Vision, 3(2), p.21.
The discussion is required for the future direction of the research in this area especially as ageing is also concerned.
Author Response
Dear Reviewer,
Thank you for your helpful comments that have helped to improve the manuscript. Below are our responses to your comments. For your convenience, we have copied the changed sections here, and also in the manuscript where the revised sections are highlighted in yellow.
Reviewer Comment 1:
The study provides evidence on robustness of bilateral eye movements for memory retrieval. There should be more discussion on why this method may or may not work on different people. For example, authors could refer to the following studies explaining their insights on how future work could enhance the memory retrieval process by adopting multimodal stimuli rather than only oculomotor involvement:
Marandi, R.Z. and Sabzpoushan, S.H., 2014. Using eye movement analysis to study auditory effects on visual memory recall. Basic and clinical neuroscience, 5(1), p.55.
or to what extent the lack of evidence could be related to the type of task for example
Leer, A., Engelhard, I.M., Lenaert, B., Struyf, D., Vervliet, B. and Hermans, D., 2017. Eye movement during recall reduces objective memory performance: An extended replication. Behaviour Research and Therapy, 92, pp.94-105.
Marandi, R.Z. and Gazerani, P., 2019. Aging and eye tracking: in the quest for objective biomarkers. Future Neurology, 14(4), p.FNL33.
Wynn, J.S., Shen, K. and Ryan, J.D., 2019. Eye movements actively reinstate spatiotemporal mnemonic content. Vision, 3(2), p.21.
The discussion is required for the future direction of the research in this area especially as ageing is also concerned.
Author Response:
Thank you for bringing our attention to this relevant literature. We have added the papers to the manuscript alongside the following paragraph expanding our discussion section based on your feedback.
“It should be considered that certain individuals may be more susceptible to enhancement effects than others and methodologies that look at enhancement effects on an individual level may show more promise [41]. It is clear from the literature that the effect is more evident in strongly right-handed individuals [10] and it is possible that other unknown characteristics may mediate the effect. Further, to date, research has not examined the potential enhancement effect on memory retrial when employing multimodal stimuli rather not solely oculomotor involvement. Marandi et al [41] demonstrated that auditory input potentially improves the visual recall process. Combining enhancement effects (e.g. oculomotor and auditory techniques) in a multimodal stimuli approach may result in greater enhancement effects on memory retrieval creating a more robust and replicable effect. Across the literature, multiple tasks and stimulus types have been employed when assessing the enhancement effects of eye movements on cognition with mixed results [10,13, 42, 43]. Future research should examine enhancement effects on memory recall and retrieval by employing a multi-modal stimuli approach and examining the effect of stimulus variables, such as emotional valence. Additional research is also required to examine the SIRE effect on multiple task types and stimuli due to the vast about of literature employing different methodologies.”
Thank you again for your helpful comments aiding us to improve the manuscript.
Yours Sincerely,
Megan Polden and Professor Trevor J. Crawford
